# Outcomes of a social media campaign to promote COVID-19 vaccination in Nigeria

W. Douglas Evans[1]*, Jeffrey B. Bingenheimer[1], Michael Long[1], Khadidiatou Ndiaye[1], Dante Donati[2], Nandan M. Rao[3], Selinam Akaba[1], Ifeanyi Nsofor[4], Sohail Agha[5]

1 Milken Institute School of Public Health, The George Washington University, Washington, DC, United States of America, 2 School of Business, Columbia University, New York, New York, United States of America, 3 Virtual Lab LLC, Corvallis, Oregon, United States of America, 4 Family Health Initiative, Abuja, Nigeria, 5 Global Health Visions, Seattle, Washington, United States of America

* wdevans@gwu.edu

**Data Availability Statement:** *****PA AT ACCEPT: Please follow up with authors for repository information available at Accept***** PLEASE NOTE THAT WE STATED WE WILL PROVIDE ACCESS TO

## Abstract

The COVID-19 pandemic has been an historic challenge to public health and behavior change programs. In low -and middle-income countries (LMICs) such as Nigeria, there have been challenges in promoting vaccination. Vaccine hesitancy and social norms related to vaccination may be important factors in promoting or inhibiting not only COVID vaccination, but other routine vaccinations as well. The aim of this study was to conduct a national-level quasi-experimental evaluation of a social media based COVID-19 vaccination promotion campaign in Nigeria run in 2022. We followed a longitudinal cohort of Nigerians (at baseline) drawn from all 37 states in Nigeria over a 10-month period. This was done at 3 time points to evaluate psychosocial predictors of vaccination and vaccination outcomes following a theory of change based on Diffusion of Innovations, Social Norms Theory, and the Motivation, Opportunity, Ability (MOA) Framework. In a quasi-experimental design, participants in 6 Nigerian states where the social media campaign was run (treatment) were compared to participants from non-treatment states. This study highlights new social media-based data collection techniques. The study found that vaccination rates increased in treatment states compared to non-treatment states, and that these effects were strongest between baseline and first follow up (December 2021 to March 2022). We also found that more pro-vaccination social norms at one time point are associated with higher vaccination rates at a later time point. Social media campaigns are a promising approach to increasing vaccination at scale in LMICs, and social norms are an important factor in promoting vaccination, which is consistent with the Social Norms Theory. We describe implications for future vaccination campaigns and identify future research priorities in this area.

## Introduction

The COVID-19 pandemic has led to over seven million deaths worldwide as of June 2023, and has been a monumental challenge for medical and public health systems worldwide. The introduction of COVID-19 vaccines in December 2020, and rollout of vaccination programs in the following years have been critical in controlling the pandemic, but have also led to the

OUR DATA UPON REQUEST, NOT THAT WE WILL
PROVIDE REPOSITORY INFORMATION. THUS WE
WILL NOT BE PROVIDING ACCESSION NUMBERS
OR DOIs AT THIS TIME.

**Funding:** The study was funded through INV-
033413 from the Gates Foundation. We also state
that the project officer who was an employee of the
funder at the time this project was initiated, Dr.
Sohail Agha, participated in discussions about the
study design and data collection. At the time of this
manuscript preparation, Dr. Agha was no longer
with the funder. Dr. Agha did participate in
development of the manuscript as stated in the
author roles section.

**Competing interests:** The authors have declared
that no competing interests exist.

challenge of vaccine hesitancy [1, 2]. The existence of mis- and disinformation, especially
online, has created a major barrier to vaccine acceptance and uptake [3]. Thus, there is need to
create demand and to understand the factors that promote and inhibit vaccine uptake in
groups that are either uninformed, vaccine hesitant, or actively resistant to COVID-19
vaccination.

COVID-19 is an historic global health challenge, and both supply and demand for vaccina-
tion in Low- and Middle-Income Countries (LMICs) is a major issue in controlling the pan-
demic worldwide. Specific Issues in LMICs include the 5 Cs of vaccine hesitancy (confidence,
complacency, convenience, communications, and context), social norms around vaccination,
mis- and disinformation, among other challenges [4]. Additionally, variables such as gender,
education levels, socio-economic status, and endorsement by trusted actors such as Health
Care Providers (HCPs) have been found to affect vaccine hesitancy and uptake in LMICs [5].

Positive role models who promote a pro-vaccination norm, such as HCPs, business leaders,
community and religious leaders, government officials, celebrities, and others are an impor-
tant potential source of social influence to promote vaccination [6]. Demonstrating the bene-
fits of vaccination and overcoming false and misleading information (e.g., about negative
health effects of vaccination, such as infertility) through influential messaging is a potentially
powerful avenue to increase vaccine uptake.

Vaccine hesitancy among important potential role models such as HCPs may be a major
barrier to the uptake of COVID-19 vaccines in large LMICs such as Nigeria. This is a focus on
vaccine uptake efforts especially given the observed role of mis- and disinformation about
COVID-19 vaccination in many countries during the pandemic [7]. HCPs have been identi-
fied by the Nigerian government as a key target population for COVID-19 vaccinations. As
HCPs will be responsible for the rollout of vaccines in the general population of Nigeria, their
lack of adoption of COVID-19 vaccines may hamper the rollout of the national campaign.
Studies in high-income countries (HICs) show that vaccinated HCPs are more likely to be suc-
cessful in persuading communities to receive vaccines. There is also some concern that, if
HCPs exhibit vaccine hesitancy, vaccines allocated to them may get diverted to other popula-
tions. By modeling vaccine acceptance behavior, HCPs are well-positioned to influence norms
of behavior amongst their colleagues. There is some evidence from HICs on effective interven-
tions to reduce vaccine hesitancy, but much more research is needed on effective strategies
and messages [8–10].

The present study aims to evaluate the effects of a social media campaign to promote
COVID-19 vaccination in Nigeria. The campaign was based on a Theory of Change (ToC)
that posited that promoting positive social norms in favor of vaccination, reducing vaccine
hesitancy, and promoting pro-vaccination Motivation, Opportunity, and Ability (MOA) to
be vaccinated would collectively lead to higher vaccination rates [11]. The ToC for this
campaign was based on Diffusion of Innovations, Social Norms Theory, MOA framework
[12–14]. The ToC specified primary and secondary outcomes in terms of COVID-19 vacci-
nation, vaccine hesitancy, and the psychological and population-based processes by which
changes in these outcomes occur. This ToC was designed as a basis for a social media cam-
paign in Nigeria using the Facebook and Instagram platforms, which have large followings
in the country. The ToC hypothesized a set of mediating variables, which included social
norms, role modeling, and cultural beliefs that are theoretically related to vaccine intentions
and vaccination behavior [15].

This study reports on findings from a national evaluation of the campaign in all 37 states in
Nigeria using a quasi-experimental design. We tested two primary hypotheses (H). First, H1:
Exposure to the social media campaign in treatment states will produce higher rates of
COVID-19 vaccination compared to non-treatment states. Second, H2: Lower levels of vaccine

hesitancy at earlier time points will be associated with higher vaccination rates at later time points, and similarly higher levels of pro-vaccination social norms at earlier time points will be associated with higher vaccination rates at later time points.

## Methods

### Design and intervention

The current study was part of a larger evaluation of the impact of social media campaigns to promote COVID-19 vaccination among HCPs and others in their social environment in Nigeria. The evaluation employed mixed methods and comprised of a quantitative study conducted through social media-based surveys, a qualitative study of stakeholders, and a cost effectiveness study. The qualitative and cost effectiveness study findings are reported elsewhere. Within the quantitative study, we collected baseline data from participants across Nigeria recruited through a Facebook-based survey, protocol described below, in December 2021, March-April 2022, and October 2022.

The campaign was designed and delivered by a separate team of designers and local organizations in Nigeria who created social media content and delivered it on Facebook and Instagram. Local organizations included healthcare organizations, businesses, religious groups, and similar prominent leaders with a substantial, existing social media presence and network. Content consisted of graphics, videos, and other social media materials such as GIF (graphic interface format) images. The campaign was run in two phases, from January to February 2022, and from May to September 2022, corresponding to the time periods just after baseline, before the first follow up, the period after the first follow up, and just before the second (endline) follow up.

The evaluation was reviewed and approved by The George Washington University Institutional Research Board (IRB) and by the National Health Research Ethics Committee (NHREC) in Nigeria. Consent was obtained by providing a consent statement to participants in the social media chatbot (described below). Participants who clicked to continue the survey after reading the statement consented to participate.

Vaccine hesitancy occurs along a continuum between full acceptance, including high demand for vaccines, and outright refusal of some or all vaccines. It is widely accepted that communications campaigns are best used to target people in the center of a spectrum—the 'persuadable middle'. The campaign therefore attempted to reach HCPs and their communities who were unsure or delaying getting vaccinated.

Using the previously described ToC as a basis, the campaign designers iterated a series of social media content focused on promoting pro-vaccination social norms and reducing vaccine hesitancy using social influencers (individuals such as local celebrities, other HCPs, religious and business leaders, and other individuals who would be respected and trusted by the audience). The campaign ToC was premised on the idea that a social media campaign that achieved high levels of reach and audience engagement would be able to reduce vaccine hesitancy, increase pro-vaccination social norms, and related attitudes and beliefs, thus increase COVID-19 vaccination uptake.

### Measures

The focal independent variable in this study was treatment arm, meaning whether the respondent's Facebook account was assigned to a campaign state or a comparison state. We constructed a treatment state indicator variable on which respondents from campaign states (Anambra, Bauchi, Lagos, Niger, Rivers, or Sokoto) received a code of 1, and those from any of

the remaining states or the Federal Capital Territory received a code of 0. The campaign states were selected by the campaign design team.

The primary endpoint in this study is COVID-19 vaccination uptake. This was assessed using a single questionnaire item which read, "Have you received a COVID-19 vaccine?" There were four response options: "Yes, a single-dose vaccine"; "Yes, the first dose of a two-dose regimen"; "Yes, both doses of a two-dose regimen"; and "No". Due to changes over the course of the study in the availability of different types of COVID-19 vaccines in Nigeria as well as recommendations for receiving boosters, we combined the first three responses into a single vaccinated category (coded 1), while those who answered "No" were coded 0 for unvaccinated.

This study also had two secondary endpoints. The first was vaccine hesitancy, an index defined as the mean of coded responses to five questionnaire items reflecting the Five Cs framework and adapted from [16]. The items read, "I am confident that COVID-19 vaccines are safe and effective," "Vaccination against COVID-19 is unnecessary," "Everyday stress prevents me from getting a COVID-19 vaccine," "When I think about getting vaccinated against COVID-19, I weigh the benefits and risks to make the best decision possible," and "When everyone is vaccinated against COVID-19, I don't have to get vaccinated too." These were intended as measures of confidence, complacency, convenience, calculation, and collective responsibility, respectively. For each of these items, respondents were asked to select one of the following five response options: "Strongly agree," "Agree," "Neither agree nor disagree," "Disagree," and "Strongly disagree." We coded these from 1 to 5 or 5 to 1 depending on the wording of the item (i.e., whether agreement reflected lower or higher levels of hesitancy). The resulting index had low internal consistency (Cronbach's a = 0.33 at baseline), but this is not surprising because the items represent diverse sources of hesitancy rather than serving as indicators of a single, unidimensional underlying construct. Because it is defined as the mean of component items, our vaccine hesitancy index has a plausible range of 1 to 5.

The study's other secondary endpoint was pro-vaccination social norms, a scale score defined as the mean of coded responses to five questionnaire items adapted from previously validated measures [17]. The first two items read, "Your friends think it is important for everyone to get a COVID-19 vaccine" and "Your family members think it is important for everyone to get a COVID-19 vaccine." The response options for each were "Strongly disagree," "Disagree," "Neither agree nor disagree," "Agree," and "Strongly agree," and we coded these 1 to 5. The remaining three questionnaire items assessing social norms read, "Of the people close to you, what proportion of them would want you to get the COVID-19 vaccine?", "How many people in Nigeria do you think will get the COVID-19 vaccine when it becomes available?", and "How many people who work in healthcare in Nigeria do you think will get the COVID-19 vaccine when it becomes available?" The response options for these were, "A few (1–20%)," "Some (21–40%)," "Many (41–60%)," "Most (61–80%)," and "Nearly all or all (81–100%)." We coded these 1 to 5, respectively. The content of these items reflects two key elements in the theory of social norms and the application thereof to health promotion interventions [16]: the distinction between descriptive norms (perceptions of what others do) and injunctive norms (perceptions of what others think one should do) [18, 19], and the potential relevance of multiple reference groups for any given health-related behavior [17, 20]. The first two items capture injunctive norms, while the third, fourth, and fifth capture descriptive norms. And together, the five items capture five different reference groups. The resulting scale, defined as the mean of the component items, had a plausible range of 1 to 5 and a moderately high internal consistence (Cronbach's a = 0.74 at baseline).

## Data collection

Survey data for the current study were collected on a social media-based research platform called Virtual Lab (https://vlab.digital/) [21]. The survey instrument was designed to measure the previously mentioned outcomes, as well as key elements in the campaign ToC, described earlier.

Virtual Lab is an open-source software platform that uses targeted digital advertising to recruit a custom sample of respondents. In this case, the study was stratified by health care worker status, and we sought to obtain 50% of our total sample in this group with the rest being the general Nigerian population. Sample size estimates aimed to detect a change in COVID-19 vaccination rates among HCPs and the general population. To be eligible, targeted participants had to be 18 years of age or older, have a Facebook account, and they received recruitment advertising in their live feed promoting a study on COVID-19 vaccination. Facebook is the 2nd leading social media platform in Nigeria (after WhatsApp), and mobile phone use in the country is the highest in Africa, enabling nationwide data collection [22]. By clicking through the ad, potential participants then engaged with a chatbot that used Facebook Messenger (FM) to send reply chats. A series of FM chats then screened and offered an opportunity for participants to provide informed consent statement following the IRB-approved protocol. Screening questions included age and health care worker status and job role, if applicable. After reading the consent statement, agreement to proceed with the survey represented consent. Eligible participants were offered an incentive of 400 Naira (about $1USD) in mobile phone credits to complete a questionnaire consisting of 40 items, each delivered as an individual FM message. Data were captured and stored in a secure Virtual Lab database.

Individuals who consented to participate were sent questionnaire items one at a time via the chatbot application. For most items, participants were prompted simply to tap on the response option of their choice to answer that item. After answering each item, participants received the next item within a few seconds, and this process was repeated through the end of the questionnaire. Responses were collected in an electronic database and exported as a.csv file for analysis.

## Data analysis

For our primary endpoint of COVID-19 vaccine uptake, we first obtained frequencies and percentages at each wave by study arm. Then, for vaccination status at first and second follow-up, we ran two linear regression models, one with the treatment state indicator as the sole independent variable, and the other with the following control variables: age group, gender, education, religion, and occupation, all assessed at baseline. We used clustered standard errors for statistical inference in these models to account for the nesting of participants within states.

Similarly, for our two secondary endpoints of vaccine hesitancy and pro-vaccination social norms, we obtained descriptive statistics at each wave by study arm, and then used linear regression to test for differences between treatment and comparison state residents. For these outcomes, we included baseline levels of the outcome variable as an additional control variable in the adjusted models. As with our analysis of vaccination status, we used clustered standard errors to account for the nesting of participants within states.

We next conducted logistic regression analysis to test two hypotheses from the campaign's theory of change: that vaccine hesitancy decreases vaccine uptake, and that pro-vaccination social norms increase vaccination uptake. We did this first using baseline levels of vaccine hesitancy and social norms to predict vaccination uptake between baseline and first follow-up; and then using vaccine hesitancy and social norms from first follow-up to predict vaccination uptake between first and second follow-ups among those who were not yet vaccinated at first

follow-up. For each pair of time periods, we ran four logistic regression models: one with vaccine hesitancy as the sole independent variable, the second with social norms as the sole independent variable, the third with both vaccine hesitancy and social norms as independent variables, and the fourth with controls for age group, gender, education, religion, and occupation. As before, we used clustered standard errors to correct standard errors and p-values for the nesting of participants within states.

Next, given the high level of attrition over the course of the study, we conducted an analysis of patterns of attrition. We created a 0-or-1 indicator variable for attrition before first follow-up and attrition before second follow-up. We cross-tabulated each of these with our treatment state indicator variable, age group, gender, education, religion, and occupation. We also created categorical versions of our baseline vaccine hesitancy index and social norms variables, with scores between 1 and 2 categorized as "lowest," scores between 2 and 3 categorized as "low," scores between 3 and 4 categorized as "high," and scores between 4 and 5 categorized as "highest." We cross-tabulated these with our attrition indicators, just as we did for the treatment arm and the socio-demographic variables. For each potential determinant of attrition, we also estimated a logistic regression model (also using clustered standard errors) to determine whether that variable's association with attrition was statistically significant. We also tested for differential attrition across arms using logistic regression models with the treatment state indicator variable, one of the baseline variables (age, gender), and the treatment indicator by baseline variable interaction terms. We used likelihood ratio tests of nested models (with versus without the interaction terms) to test whether the pattern of attrition was differential for each baseline variable.

Finally, as robustness checks, we conducted two alternative versions of our main analyses. In the first, we used carry-forward imputation for missing values of vaccination status, vaccine hesitancy, and pro-vaccination social norms at first and second follow-ups. Because being vaccinated at enrollment was an exclusion criterion, this meant that everyone who was lost to follow-up was assumed to still be unvaccinated at those follow-up time points in these analyses. We repeated the primary analyses (described in the first two paragraphs of this section) using the dataset with carry-forward imputation for missing values. Next, we used multiple imputation by chained equations [23, 24] to create 50 completed datasets. Input variables for the imputations included all baseline demographic variables as well as the baseline vaccine hesitancy index and pro-vaccination social norms scale. We then carried out analyses analogous to our primary ones using these imputed datasets and obtained aggregated point estimates and standard errors following the formula derived by Little and Rubin [25]. All data management and analysis were carried out in Stata version 17 [26]. Clustered standard errors were obtained by adding the option *vce(cluster state)* to all linear and logistic regression commands. For the multiple imputations and the analyses of the imputed datasets we used Stata's *mi impute chained* command and regression commands with the *mi estimate* prefix.

## Results

Fig 1 is the CONSORT diagram that shows the flow of (potential) participants into and through the study. A total of 10,965 potential participants received links to the Facebook Messenger questionnaire and began answering screening questions. Over 82% did not meet the eligibility criteria, with already having received a vaccination against COVID-19 accounting for the majority of exclusions, followed by not being in the "persuadable middle" in terms of intention to get vaccinated. The persuadable middle was defined as those who responded that they were not already vaccinated were asked, "Would you take a COVID-19 vaccine that is approved for use in Nigeria if offered to you?" Those who responded "Definitely" or

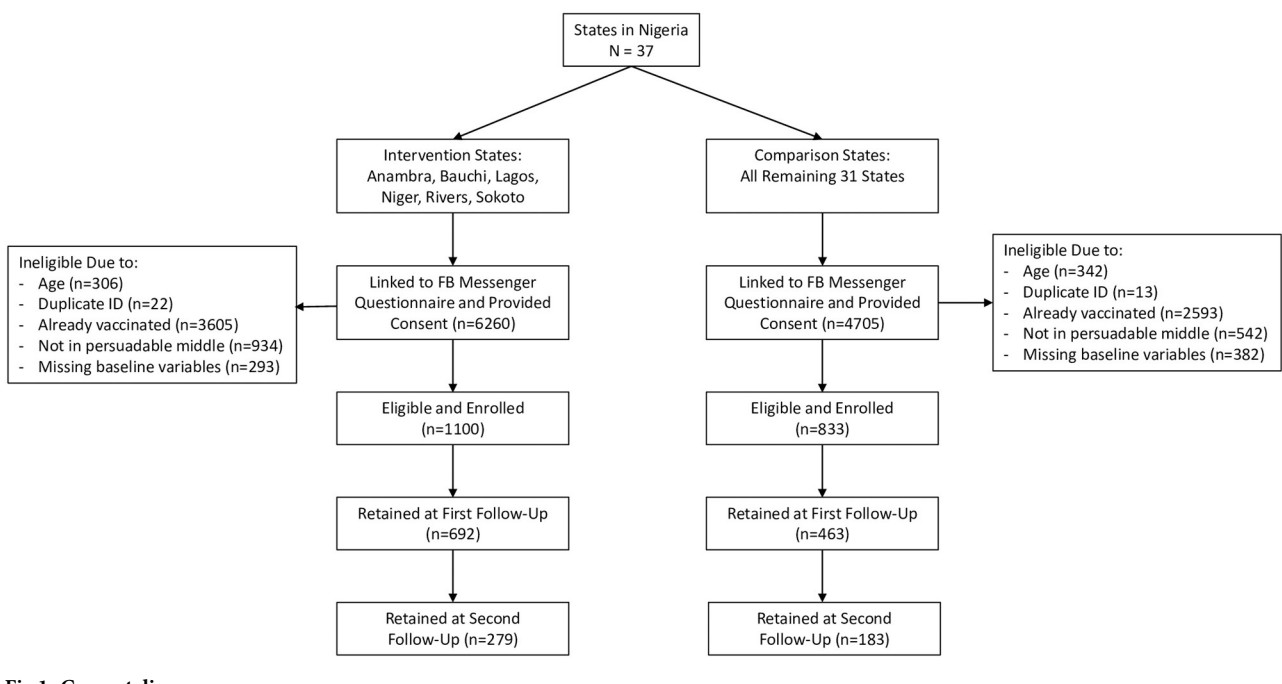

**Fig 1. Consort diagram.**

"Definitely not" were not in the persuadable middle. Additional exclusions included missing baseline variables, being below age 18, and having a duplicate ID number (which occurred when someone from a single Facebook account linked to the questionnaire more than once). We thus enrolled 1,100 participants from the six campaign (treatment) states and 833 from the 31 comparison states. Of these, 1,155 (59.8%) completed the first follow-up questionnaire, and 462 (23.9%) completed the second follow-up questionnaire.

Table 1 shows the composition of the sample at baseline. The sample was young, with over 60% being under age 30. The majority were men, and more than half had earned a secondary school diploma or higher. The most common religious affiliation was non-Catholic Christian, followed by Muslim, Roman Catholic, and others. Just over two-thirds were not HCPs, and those who were HCPs were distributed across a variety of specific occupations. Some differences in the socio-demographic composition from participants from treatment versus comparison states are evident. S1 Table in S1 File shows the distribution of treatment state participants across the six treatment states, and of comparison state participants across the 31 comparison states. Notably, one state, namely Lagos, contributed to the vast majority of treatment state participants, whereas comparison state participants were somewhat more widely distributed across the comparison states.

The main findings for primary and secondary endpoints at the first and second follow-up questionnaires are presented in Table 2. At first follow-up, 31.7% of retained participants from treatment states, compared to 25.3% of retained participants from comparison states, for a crude difference of 6.4 percentage points (p = 0.045) and an adjusted difference of 7.8 percentage points (p = 0.020). At second follow-up, these figures increased to 44.1% and 35.0% for retained treatment and comparison state participants, respectively. The crude and adjusted differences at second follow-up in percentage point terms were 9.1 and 11.0, but these were not statistically distinguishable from the null (p = 0.076 for the crude and p = 0.068 for the adjusted difference), although these changes approach the level of statistical significance. Vaccine

**Table 1. Sociodemographic composition of the baseline sample, by type of state of residence (Treatment versus comparison).**

| | Treatment State Participants (n = 1100) | Comparison State Participants (n = 833) | All Participants (n = 1933) |
|---|---|---|---|
| | n (%) | n (%) | n (%) |
| Age | | | |
| 18 to 29 years | 792 (72.0) | 444 (53.3) | 1236 (63.9) |
| 30 to 39 years | 236 (21.5) | 282 (33.9) | 518 (26.8) |
| 40 to 49 years | 49 (4.5) | 71 (8.5) | 120 (6.2) |
| 50 to 59 years | 16 (1.5) | 24 (2.9) | 40 (2.1) |
| 60+ years | 7 (0.6) | 12 (1.4) | 19 (1.0) |
| Gender | | | |
| Man | 786 (71.5) | 411 (49.3) | 1197 (61.9) |
| Woman | 285 (25.9) | 393 (47.2) | 678 (35.1) |
| Prefer Not to Say | 29 (2.6) | 29 (3.5) | 58 (3.0) |
| Education | | | |
| Primary | 11 (1.0) | 4 (0.5) | 15 (0.8) |
| Secondary | 292 (26.6) | 95 (11.4) | 387 (20.0) |
| Diploma | 228 (20.7) | 163 (19.6) | 391 (20.2) |
| Bachelors | 466 (42.4) | 453 (54.4) | 919 (47.5) |
| Masters | 50 (4.6) | 68 (8.2) | 118 (6.1) |
| Doctorate | 12 (1.1) | 5 (0.6) | 17 (0.9) |
| Other | 41 (3.7) | 45 (5.4) | 86 (4.5) |
| Religion | | | |
| Other Christian | 576 (52.4) | 493 (59.2) | 1069 (55.3) |
| Muslim | 261 (23.7) | 177 (21.3) | 438 (22.7) |
| Catholic | 212 (19.3) | 137 (16.5) | 349 (18.1) |
| Other | 35 (3.2) | 21 (2.5) | 56 (2.9) |
| Traditionalist | 16 (1.5) | 5 (0.6) | 21 (1.1) |
| Employment/Occupation | | | |
| Nurse/Midwife | 47 (4.3) | 66 (7.9) | 113 (5.9) |
| Laboratory Staff | 50 (4.6) | 54 (6.5) | 104 (5.4) |
| Community Health Worker | 37 (3.4) | 52 (6.2) | 89 (4.6) |
| Pharmacist | 29 (2.6) | 44 (5.3) | 73 (3.8) |
| Medical Doctor | 16 (1.5) | 31 (3.7) | 47 (2.4) |
| PPMV/Chemist | 6 (0.6) | 23 (2.8) | 29 (1.5) |
| Other Public Health | 57 (5.2) | 111 (13.3) | 168 (8.7) |
| Not a Health Sector Worker | 858 (78.0) | 452 (54.3) | 1310 (67.8) |

hesitancy scale scores decreased slightly on average among treatment state participants, but also decreased among participants from comparison states, and neither the crude nor the adjusted differences at either first or second follow-up were statistically distinguishable from the null. Scores on the pro-vaccination social norms scale increased very slightly between waves in participants from treatment and control states, but none of the crude or adjusted differences were statistically significant. S2 Table in S1 File provides a more detailed look at vaccine hesitancy and pro-vaccination social norms at the item rather than scale level, where crude and adjusted differences at both follow-up time points were almost universally statistically indistinguishable from the null.

Table 3 presents results of tests of hypotheses that vaccine hesitancy at one point in time would be negatively associated with uptake of COVID-19 vaccination before the next time

**Table 2. Levels of vaccination uptake, vaccine hesitancy, and pro-vaccination norms, and crude and adjusted differences at first and second follow-ups.**

| Levels | Vaccinated Percent and (Count) | | Vaccine Hesitancy Scale Mean and (SD) | | Pro-Vaccination Social Norms Scale Mean and (SD) | |
|---|---|---|---|---|---|---|
| | Treatment | Comparison | Treatment | Comparison | Treatment | Comparison |
| Baseline | 0.0% (0/1100) | 0.0% (0/833) | 2.87 (0.54) | 2.92 (0.53) | 2.98 (0.78) | 2.85 (0.76) |
| First Follow-Up | 31.7% (219/692) | 25.3% (117/463) | 2.84 (0.54) | 2.83 (0.55) | 3.05 (0.79) | 3.05 (0.79) |
| Second Follow-Up | 44.1% (123/279) | 35.0% (64/183) | 2.79 (0.57) | 2.76 (0.58) | 3.09 (0.89) | 3.12 (0.77) |
| Differences (T–C) | Estimate | (p-value) | Estimate | (p-value) | Estimate | (p-value) |
| Crude at First Follow-Up | 6.4 | (0.045) | 0.01 | (0.688) | 0.00 | (0.950) |
| Adjusted at First Follow-Up | 7.8 | (0.020) | -0.00 | (0.961) | 0.02 | (0.696) |
| Crude at Second Follow-Up | 9.1 | (0.076) | 0.03 | (0.370) | -0.02 | (0.822) |
| Adjusted at Second Follow-Up | 11.0 | (0.068) | -0.00 | (0.948) | -0.00 | (0.956) |

point, and that pro-vaccination social norms would be positively associated with vaccination uptake over the same periods. Across all of these models, none of the crude or adjusted odds ratio for the relationship between vaccine hesitancy and vaccination uptake are statistically significantly different from the null, although most are in the hypothesized negative direction. In contrast, all of the crude and adjusted odds ratios for pro-vaccination social norms are statistically significant, with point estimates that suggest that a unit increase in pro-vaccination social norms increases the odds of being vaccinated between waves by 50 to 60 percent.

Table 4 presents results of a detailed analysis of patterns of attrition between baseline and first follow-up, and between baseline and second follow-up. Overall levels of loss to follow-up were similar among participants in treatment and comparison states. But several sociodemographic variables were statistically significantly associated with being lost to follow-up rather than retained. For example, participants in the oldest age category were substantially more likely to be lost to follow-up than those in younger age categories, and medical doctors and pharmacists were more likely than participants in other health sector and non-health fields to be lost to follow-up. Loss to follow-up also differed in relation to baseline vaccine hesitancy, with participants at the lowest and highest levels of the vaccine hesitancy scale being more likely than those in the middle ranges of vaccine hesitancy. Moreover, several of the study arm by background variable interactions were statistically significant, suggesting that the pattern of

**Table 3. Logistic regression models with vaccine hesitancy and pro-vaccination social norms predicting vaccination uptake between baseline and first follow-up, and between first follow-up and second follow-up.**

| | Hesitancy and Norms at Baseline Predicting Uptake before First Follow-Up among Those Unvaccinated at Baseline (n = 1155) | | | | | | | |
|---|---|---|---|---|---|---|---|---|
| | Model 1 | | Model 2 | | Model 3 | | Model 4 | |
| | OR | (p-value) | OR | (p-value) | AOR | (p-value) | AOR | p-value |
| Vaccine Hesitancy | 0.88 | (0.272) | | | 1.01 | (0.935) | 1.04 | (0.704) |
| Pro-Vaccine Social Norms | | | 1.53 | (0.000) | 1.53 | (0.000) | 1.53 | (0.000) |
| | Hesitancy and Norms at First Follow-Up Predicting Uptake before Second Follow-Up among Those Unvaccinated at First Follow-Up (n = 329) | | | | | | | |
| | Model 1 | | Model 2 | | Model 3 | | Model 4 | |
| | OR | (p-value) | OR | (p-value) | AOR | (p-value) | AOR | p-value |
| Vaccine Hesitancy | 0.78 | (0.246) | | | 0.87 | (0.566) | 0.85 | (0.511) |
| Pro-Vaccine Social Norms | | | 1.59 | (0.001) | 1.56 | (0.002) | 1.57 | (0.005) |

Note: OR = Odds ratio and AOR = Adjusted odds ratio. Model 4 in both panels include controls for age, education, employment, gender, and religion.

**Table 4. Patterns of Loss-to-Follow-Up (LTFU) between baseline and first and second follow-ups.**

| | N at Baseline | Between Baseline and First Follow-up | | | | | Between Baseline and Second Follow-up | | | | |
|---|---|---|---|---|---|---|---|---|---|---|---|
| | | LTFU % | OR | p-value | Omnibus p-value | Interaction p-value | LTFU % | OR | p-value | Omnibus p-value | Interaction p-value |
| Study Condition | | | | | | | | | | | |
| Control State | 833 | 44.4 | 1.00 | Ref | 0.528 | | 78.0 | 1.00 | Ref | 0.509 | |
| Treatment State | 1100 | 37.1 | 0.74 | 0.528 | | | 74.6 | 0.83 | 0.509 | | |
| Age Group | | | | | | | | | | | |
| 18 to 29 years | 1236 | 40.9 | 1.00 | Ref | 0.044 | 0.159 | 75.7 | 1.00 | Ref | 0.981 | 0.505 |
| 30 to 39 years | 518 | 40.0 | 0.96 | 0.566 | | | 76.3 | 1.03 | 0.814 | | |
| 40 to 49 years | 120 | 33.3 | 0.72 | 0.222 | | | 75.0 | 0.96 | 0.878 | | |
| 50 to 59 years | 40 | 30.0 | 0.62 | 0.150 | | | 77.5 | 1.10 | 0.776 | | |
| 60+ years | 19 | 68.4 | 3.13 | 0.037 | | | 100.0 | | | | |
| Gender | | | | | | | | | | | |
| Male | 1197 | 39.5 | 1.00 | Ref | 0.341 | 0.040 | 74.4 | 1.00 | Ref | 0.171 | 0.087 |
| Female | 678 | 41.0 | 1.06 | 0.767 | | | 78.8 | 1.28 | 0.105 | | |
| Prefer Not to Say | 58 | 46.6 | 1.33 | 0.248 | | | 81.0 | 1.47 | 0.181 | | |
| Education | | | | | | | | | | | |
| Primary | 15 | 46.7 | 1.00 | Ref | 0.000 | 0.061 | 80.0 | 1.00 | Ref | 0.000 | 0.013 |
| Secondary | 387 | 38.0 | 0.70 | 0.151 | | | 81.1 | 1.08 | 0.806 | | |
| Diploma | 391 | 36.3 | 0.65 | 0.074 | | | 74.2 | 0.72 | 0.300 | | |
| Bachelors | 919 | 41.7 | 0.82 | 0.577 | | | 74.4 | 0.73 | 0.354 | | |
| Masters | 118 | 44.9 | 0.93 | 0.843 | | | 80.5 | 1.03 | 0.921 | | |
| Doctorate | 17 | 64.7 | 2.10 | 0.114 | | | 82.4 | 1.17 | 0.748 | | |
| Other | 86 | 40.7 | 0.78 | 0.386 | | | 72.1 | 0.65 | 0.311 | | |
| Employment | | | | | | | | | | | |
| Community Health Worker | 89 | 39.3 | 1.00 | Ref | 0.000 | 0.011 | 68.5 | 1.00 | Ref | 0.083 | 0.006 |
| Not in Health Sector | 1310 | 39.9 | 1.02 | 0.920 | | | 76.7 | 1.51 | 0.064 | | |
| Other Public Health | 168 | 32.7 | 0.75 | 0.323 | | | 71.4 | 1.15 | 0.604 | | |
| Nurse/Midwife | 113 | 40.7 | 1.05 | 0.824 | | | 82.3 | 2.13 | 0.010 | | |
| Laboratory Staff | 104 | 36.5 | 0.88 | 0.665 | | | 74.0 | 1.31 | 0.285 | | |
| Pharmacist | 73 | 57.5 | 2.09 | 0.007 | | | 76.7 | 1.51 | 0.164 | | |
| Medical Doctor | 47 | 63.8 | 2.72 | 0.007 | | | 83.0 | 2.24 | 0.112 | | |
| PPMV/Chemist | 29 | 34.5 | 0.81 | 0.560 | | | 69.0 | 1.02 | 0.970 | | |
| Religion | | | | | | | | | | | |
| Catholic | 349 | 45.0 | 1.00 | Ref | 0.000 | 0.026 | 79.9 | 1.00 | Ref | 0.060 | 0.001 |
| Other Christian | 1069 | 41.7 | 0.88 | 0.233 | | | 76.2 | 0.80 | 0.115 | | |
| Muslim | 438 | 31.5 | 0.56 | 0.080 | | | 72.2 | 0.65 | 0.034 | | |
| Other | 56 | 41.1 | 0.85 | 0.469 | | | 82.1 | 1.15 | 0.536 | | |
| Traditionalist | 21 | 66.7 | 2.45 | 0.024 | | | 76.2 | 0.80 | 0.673 | | |
| Baseline Hesitancy | | | | | | | | | | | |
| Lowest | 77 | 54.6 | 1.00 | Ref | 0.000 | 0.007 | 79.2 | 1.00 | Ref | 0.105 | 0.258 |
| Low | 863 | 38.4 | 0.52 | 0.006 | | | 74.9 | 0.78 | 0.262 | | |
| High | 939 | 40.3 | 0.56 | 0.009 | | | 76.6 | 0.86 | 0.468 | | |
| Highest | 54 | 50.0 | 0.83 | 0.638 | | | 83.3 | 1.31 | 0.517 | | |
| Baseline Social Norms | | | | | | | | | | | |
| Lowest | 192 | 44.3 | 1.00 | Ref | 0.592 | 0.001 | 78.7 | 1.00 | Ref | 0.000 | 0.006 |
| Low | 750 | 39.5 | 0.82 | 0.198 | | | 72.9 | 0.73 | 0.064 | | |
| High | 767 | 40.0 | 0.84 | 0.497 | | | 79.5 | 1.05 | 0.810 | | |
| Highest | 224 | 40.2 | 0.85 | 0.609 | | | 72.8 | 0.73 | 0.082 | | |

**Table 5.** A. First Robustness Check Results: Levels of Vaccination Uptake, Vaccine Hesitancy, and Pro-Vaccination Norms, and Crude and Adjusted Differences at First and Second Follow-Ups with Simple Carry-Forward Imputation. B. Second Robustness Check Results: Levels of Vaccination Uptake, Vaccine Hesitancy, and Pro-Vaccination Norms, and Crude and Adjusted Differences at First and Second Follow-Ups with Multiple Model-Based Imputation via Chained Equations.

A

|  | Vaccinated Percent and (Count) | | Vaccine Hesitancy Scale Mean and (SD) | | Pro-Vaccination Social Norms Scale Mean and (SD) | |
|---|---|---|---|---|---|---|
| Levels | Treatment | Comparison | Treatment | Comparison | Treatment | Comparison |
| Baseline | 0.0% (0/1100) | 0.0% (0/833) | 2.87 (0.54) | 2.92 (0.53) | 2.98 (0.78) | 2.85 (0.76) |
| First Follow-Up | 19.9% (219/1100) | 14.1% (117/833) | 2.84 (0.56) | 2.88 (0.55) | 3.06 (0.79) | 2.92 (0.80) |
| Second Follow-Up | 23.5% (258/1100) | 15.9% (64/833) | 2.82 (0.56) | 2.87 (0.55) | 2.90 (0.67) | 2.95 (0.61) |
| Differences (T–C) | Estimate | (p-value) | Estimate | (p-value) | Estimate | (p-value) |
| Crude at First Follow-Up | 5.8 | (0.097) | -0.04 | (0.099) | 0.14 | (0.053) |
| Adjusted at First Follow-Up | 6.4 | (0.053) | -0.01 | (0.581) | 0.05 | (0.071) |
| Crude at Second Follow-Up | 7.6 | (0.050) | -0.05 | (0.029) | -0.05 | (0.032) |
| Adjusted at Second Follow-Up | 8.2 | (0.026) | -0.02 | (0.165) | -0.04 | (0.136) |

B

|  | Vaccinated Percent and (Count) | | Vaccine Hesitancy Scale Mean and (SD) | | Pro-Vaccination Social Norms Scale Mean and (SD) | |
|---|---|---|---|---|---|---|
| Levels | Treatment | Comparison | Treatment | Comparison | Treatment | Comparison |
| Baseline | 0.0% (0/1100) | 0.0% (0/833) | 2.87 (0.54) | 2.92 (0.53) | 2.98 (0.78) | 2.85 (0.76) |
| First Follow-Up | 31.3% (344.7/1100) | 26.0% (216.9/833) | 2.83 (0.55) | 2.84 (0.55) | 3.08 (0.80) | 3.02 (0.81) |
| Second Follow-Up | 44.1% (484.7/1100) | 39.1% (325.3/833) | 2.77 (0.59) | 2.79 (0.60) | 3.16 (0.87) | 3.09 (0.85) |
| Differences (T–C) | Estimate | (p-value) | Estimate | (p-value) | Estimate | (p-value) |
| Crude at First Follow-Up | 5.3 | (0.051) | -0.01 | (0.667) | 0.06 | (0.277) |
| Adjusted at First Follow-Up | 5.2 | (0.056) | -0.01 | (0.682) | 0.01 | (0.896) |
| Crude at Second Follow-Up | 5.0 | (0.174) | -0.02 | (0.546) | 0.07 | (0.304) |
| Adjusted at Second Follow-Up | 5.2 | (0.148) | -0.02 | (0.520) | 0.01 | (0.805) |

attrition may have been different among participants from treatment states than among those from comparison states.

Findings of sensitivity analysis intended to address the large magnitude and differential pattern of loss to follow-up are presented in Table 5. Panel A shows findings from carry-forward imputation analyses. For vaccination uptake, the point estimates are in the same direction as those presented in Table 2, with vaccination uptake being more common among participants from treatment states than from comparison states. While these differences are not statistically significant at first follow-up, they are statistically significant at second follow-up. For vaccine hesitancy index scores, crude and adjusted differences at first and second follow-ups remain small, but one–the crude difference at second follow-up–is now statistically significant and favors treatment state participants. Crude and adjusted differences in pro-vaccination social norms scale scores are likewise small, and the one difference that is statistically significant–the crude difference at second follow-up–actually favors comparison state participants. Panel B shows findings from multiple model-based imputation. None of the differences between treatment and comparison state participants are statistically significantly different from the null in these analyses, but the point estimates for vaccination uptake are in the positive direction and are close to being statistically significant in the models for vaccination uptake a first follow-up.

## Discussion

This study aimed to evaluate the effects of a large-scale social media campaign to promote COVID-19 vaccination in Nigeria using a quasi-experimental design. Given that there have

been relatively few of such large-scale evaluations of social media to promote behavior change, especially in LMICs, this research has the potential to improve the evidence base in the field [27]. The study tested a set of hypotheses based on a grounded ToC, which has been absent from many previous social media campaigns in LMICs [27].

The main overall finding confirms H1: There is evidence that the campaign promoted higher levels of vaccination in treatment states compared to comparison states. As stated earlier, both the crude and adjusted models reflect a positive effect of the campaign on retained participants in treatment states compared to control states. At the first follow-up, 31.7% of retained participants from treatment states had been vaccinated, compared to 25.3% of retained participants from comparison states who had been vaccinated, for a crude difference of 6.4 percentage points (p = 0.045) and an adjusted difference of 7.8 percentage points (p = 0.020). While the magnitude of these differences were even greater at the second follow up, there were 9.1 and 11.0 percentage points of higher vaccination among retained treatment state participants compared to comparison state participants; the results approached but did not reach the level of statistical significance, defined as $P < .05$.

It is worth noting that we had substantial loss to follow up (LTFU) by the time of the second follow up. While we did not observe differential attrition by treatment state versus comparison state, some differences in LTFU were observed between demographic characteristics, and by vaccine hesitancy status (i.e., the extreme responses were most likely to be LTFU, as compared to the middle of the hesitancy range). The substantial LTFU, while not different by study condition, may have reduced power to detect an effect of treatment on vaccination rates at the second follow up, and this should be considered in interpreting the near significant effects in the treatment states at the second time point.

The secondary analyses partially confirm H2: While we did not find that vaccine hesitancy at one time is associated with higher vaccination at a later time point, we did find evidence that more pro-vaccination social norms at one time point are associated with higher vaccination rates at a later time point. As stated earlier, none of the crude or adjusted models show a statistically significantly relationship between vaccine hesitancy and vaccination uptake, although most are in the hypothesized negative direction (i.e., higher vaccine hesitancy is associated with lower vaccination rates). However, all of the crude and adjusted odds ratios for pro-vaccination social norms are statistically significant, and the models suggest that a unit increase in pro-vaccination social norms increases the odds of being vaccinated between waves by 50 to 60 percent. This suggests that pro-vaccination normative beliefs may be an important driver of COVID-19 vaccination.

Similar overall considerations in terms of LTFU should be considered for the secondary outcomes. We observed some differences in attrition for very low and very high vaccine hesitancy respondents, and this may have affected the results for the relationship between vaccine hesitancy at an earlier time point and vaccination status at a later time point. We did not observe similar attrition differences based on social norms scale responses.

Overall, we conclude that there is evidence that the campaign produced measurable and practically meaningful positive effects on social norms and vaccination status. The fact that social norms were hypothesized as one mediator of the effects on vaccination provides support for the campaign's ToC. Further analysis of the potential mediating effects of social norms on vaccination should be conducted, and future research should examine the potential for social norms to be a mediator of COVID-19 and other vaccine uptake.

One future avenue for new interventions is to identify normative beliefs that increase vaccine hesitancy or otherwise serve as barriers to getting vaccinated, such as motivations and beliefs about opportunities and ability to get vaccinated (per the MOA framework) [28, 29]. Myths and misinformation about the effects of COVID-19 and other vaccines (e.g., that they

cause infertility) are likely part of the mosaic of normative beliefs that increase vaccine hesitancy, and should be addressed through messaging and other intervention strategies [3, 30].

Recommendations for future research include employing stronger quasi-experimental designs such as cluster randomization in future studies on social media programs to promote vaccination. While circumstances on the ground in Nigeria at the time of this project prevented such an approach, randomly assigning sufficient states to treatment and control would have reduced important limitations of this study and enhanced generalizability. Further development and evaluation of outcomes measures used here, including vaccine hesitancy and social norms, should be done in the future. Finally, evaluation of the campaign theory of change should be conducted through formal mediation analysis (Structural Equation Modeling).

One area to explore further in future research on such campaigns is the mechanisms by which social norms are changed and influence vaccination outcomes via social media. One hypothesis is that social media may create a widespread sense of engagement in behaviors and social acceptance of behaviors such as COVID-19 and other forms of vaccination [31]. The mechanisms by which social media influence vaccination social norms deserve further experimental research.

## Limitations of the current research

The design was non-randomized, and the treatment states were selected purposefully by the campaign design team, which limits the generalizability of findings. To address this, and to be reported elsewhere, we conducted a separate randomized controlled study of individuals in non-treatment states. We also experienced relatively high attrition. The social media-based sample was collected by convenience on the Facebook platform and is thus not nationally representative of Nigerians. Also, we did not have detailed information on possible contamination of participant awareness of campaign messages from other sources, such as Nigerian government vaccination campaigns running at the same time as our campaign and evaluation study. However, we also note that the intervention reported here was the only statewide social media campaign running in the treatment states during the evaluation. Finally, while we did not observe differential attrition by treatment condition, however, we did observe some differences in attrition rates by demographic and psychographic (vaccine hesitancy) status. As a result of these differences in LTFU, findings of this study should be interpreted with caution.

Finally, the evaluation will contribute both to the knowledge of digital interventions in LMICs, specifically in Nigeria, and also to interventions for HCPs in LMICs. The health care sector is growing rapidly along with the population in LMICs, especially in sub-Saharan Africa. Best practices to improve care include a focus on HCP behavior, and their role in promoting population health through role modeling of healthy behaviors such as vaccination. This project provides one of the first and largest examples of best practices in promoting healthy behaviors among HCPs in an LMIC setting. Findings from this project contribute to the evidence base and support the global agenda to promote vaccination in LMICs [32].

## Conclusions

We found that the campaign had positive effects on vaccination status and that pro-vaccination social norms are associated with higher rates of vaccination. Social media has the potential to influence vaccination rates in LMIC settings, and there is evidence to suggest that social norms are a promising mechanism to communicate pro-vaccination messages. Further research on pro-vaccination social norms messages is needed. Additionally, there is a need to understand the quantity and frequency of social media messages that are needed to increase

vaccine willingness and vaccination rates [33]. More research on dose-response effects of pro-vaccination campaigns is needed.

## Supporting information

**S1 File. Further details on the geographic distribution of the sample and item-level averages for the 5 Cs and social norms items.**
(DOCX)

**S2 File. Codebook for PLOS One dataset.**
(DOCX)

**S1 Dataset.**
(XLSX)

## Acknowledgments

The social media campaign evaluated in this study was led by M&C Saatchi, Upswell, and a consortium of Nigerian organizations that published the social media content with approval from the Nigerian government. We thank these organizations for their collaboration.

## Author Contributions

**Conceptualization:** W. Douglas Evans, Sohail Agha.

**Data curation:** W. Douglas Evans, Jeffrey B. Bingenheimer.

**Formal analysis:** W. Douglas Evans, Jeffrey B. Bingenheimer.

**Funding acquisition:** W. Douglas Evans.

**Investigation:** W. Douglas Evans, Jeffrey B. Bingenheimer.

**Methodology:** W. Douglas Evans, Jeffrey B. Bingenheimer, Michael Long, Khadidiatou Ndiaye, Dante Donati, Nandan M. Rao.

**Project administration:** W. Douglas Evans, Dante Donati, Nandan M. Rao, Ifeanyi Nsofor.

**Software:** Jeffrey B. Bingenheimer.

**Supervision:** W. Douglas Evans.

**Validation:** W. Douglas Evans, Jeffrey B. Bingenheimer.

**Visualization:** W. Douglas Evans, Jeffrey B. Bingenheimer.

**Writing – original draft:** W. Douglas Evans, Jeffrey B. Bingenheimer.

**Writing – review & editing:** W. Douglas Evans, Khadidiatou Ndiaye, Selinam Akaba, Ifeanyi Nsofor, Sohail Agha.

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
