## [Decision Letter · Decision Letter 0]

18 Jun 2023

PONE-D-23-15997Outcomes of a social media campaign to promote COVID-19 vaccination in NigeriaPLOS ONE

Dear Dr. Evans,

Thank you for submitting your manuscript to PLOS ONE. After careful consideration, we feel that it has merit but does not fully meet PLOS ONE’s publication criteria as it currently stands. Therefore, we invite you to submit a revised version of the manuscript that addresses the points raised during the review process.

We look forward to receiving your revised manuscript.

Kind regards,

Naeem Mubarak, PhD

Academic Editor

PLOS ONE

Journal Requirements:

"Yes, Sohail Agha was a project officer and participated in meetings to design the study and provided input to the manuscript. The Principal Investigator, W. Douglas Evans, made all final study decisions."

"This study was funded by the Bill & Melinda Gates Foundation (BMGF) grant INV-033413. We thank the foundation for its support of this study."

"Yes, Sohail Agha was a project officer and participated in meetings to design the study and provided input to the manuscript. The Principal Investigator, W. Douglas Evans, made all final study decisions."

7. Please upload a copy of Figure 1, to which you refer in your text on page 11. If the figure is no longer to be included as part of the submission please remove all reference to it within the text.

Reviewers' comments:

Reviewer's Responses to Questions

**Comments to the Author**

1. Is the manuscript technically sound, and do the data support the conclusions?

Reviewer #1: Yes

Reviewer #2: Partly

2. Has the statistical analysis been performed appropriately and rigorously? 

Reviewer #1: Yes

Reviewer #2: Yes

3. Have the authors made all data underlying the findings in their manuscript fully available?

Reviewer #1: Yes

Reviewer #2: Yes

4. Is the manuscript presented in an intelligible fashion and written in standard English?

Reviewer #1: Yes

Reviewer #2: Yes

5. Review Comments to the Author

Reviewer #1: Thank you for providing the opportunity to review this interesting study

The study focused on the impact of a social media campaign on COVID-19 vaccination that was premised on the principal of the theory of change (diffusion of innovations, social norm theory, and MOA framework model) among healthcare providers and the general population in Nigeria. The research evaluated the outcomes of the status of COVID-19 vaccine uptake, vaccine hesitancy, and pro-vaccination social norms among the participants recruited at 3 different time points over 10 months.

The study holds a good deal of merit for publication with following recommendations:

The introduction section does not sufficiently cover the topic mentioned in hand. Many studies can be cited from other LMIC, for instance:

1. https://doi.org/10.37723/jumdc.v12i2.624

2. https://doi.org/10.37723/jumdc.v13i3.762

The authors have targeted Facebook and Instagram to propagate social media campaigns. How this selection will ensure the generalizability of responses in the Nigerian context and not create a demographic bias? Please clarify what you mean by “the persuadable middle”

The authors have chosen a mixed method study design. However, no detail or mention of the qualitative part of the survey. No mention of the stakeholders involved. How will you justify Line 99-101: “The evaluation employed mixed methods and comprised of a quantitative study conducted through social media-based surveys, a qualitative study of stakeholders, and a cost-effectiveness study”?

The study mentions a quasi-experimental design with non-equivalent control and treatment groups. How will you justify the 6 states that you have chosen for treatment interventions? What was the rationale for not including the federal capital of Nigeria in the treatment group? Also, the terminology for mentioning the comparison or control group keeps on changing throughout the manuscript. Would have been better to keep it consistent for a better understanding

Can the authors provide explicit detail on the social media campaign content that was used to assess the outcomes? Can they provide details on the interventional strategies used? How have they ensured that the findings obtained were merely based on the impact of social media campaigns and not on the other competing exposures?

The study mentions attrition bias as the limitation. However, can the authors provide a compelling rationale for the large loss of participants after the first follow-up?

To assess the primary outcome of COVID-19 vaccine status, please clarify does the fully vaccinated status was given with or without booster shots. How have you differentiated the 5Cs of vaccine hesitancy among participants with or without booster shots?

The discussion section lacks depth. Would have been better to draw comparisons of social media influence on COVID-19 vaccine hesitancy with other LMICs. The study demands a holistic explanation of the impact of social media campaigns in terms of the primary and secondary outcomes of your study.

Reviewer #2: In this manuscript, Evans and colleagues conducted a national-level quasi-experimental evaluation of a social media based COVID-19 vaccination promotion campaign across Nigeria's 36 states and the Federal Capital Territory. The authors purposively chose 6 States in the south-south, south-east, south-west, north-east, north-west and north-central region of Nigeria as "intervention" states where they ran online campaigns and compared them to Nigeria's other 30 States and the Federal Capital Territory as "comparator States".

The study used a potentially novel methodology for data collection. It is also evident that a lot of work and collbaorative effort has gone in to ensure this study was succesfully completed. That being said, there are some important issues that I strongly recommend that the authors address to improve the overall quality of the manuscript. Ihave lsited below as major and minor issues.

MAJOR

1. I would like to address a significant concern regarding the absence of local authors in the list of authors. In recent years, the global health community has increasingly emphasized the importance of equitable and inclusive collaborations, particularly in research conducted in low- and middle-income countries. It is crucial to ensure that local researchers, scientists, and collaborators from the host country are included as authors, as their perspectives, insights, and expertise are vital in conducting research that accurately reflects the local context and addresses the specific health needs of the population. I note that in line 104, the authors state that were invloved in designing the campaign. I also note that the study has IRB from Nigeria's national primary health care agency (not "public health agency" as stated by the authors") which I assume was led by local collaborators too. To make matthers worse, your acknowledgment section only mentioned the study funder and not even a mention of a lcoal collaborator whom have been referred to in the body of the manuscript as creators of the campaign and delivered the campaigns.

Therefore, I strongly recommend that you consider including local authors who have contributed significantly to the study design, data collection, analysis, interpretation, or manuscript preparation. This would enhance the credibility and relevance of the research findings, foster meaningful partnerships, and support the broader goal of promoting local ownership and sustainability in research endeavors. I understand that there may be various reasons for the absence of local authors in the current version of the manuscript. If there are limitations or challenges in involving local authors, I encourage you to provide a clear and transparent explanation in the manuscript, outlining the efforts made to engage local collaborators and the reasons for their exclusion as authors. By addressing this issue, your study will not only comply with emerging guidelines and expectations in global health research but also contribute to the advancement of ethical research practices, respectful collaborations, and knowledge co-creation in LMIC settings.

2. The manuscript methods sections lacked information on the local context in Nigeria which would be useful in contextualizing the findings. For example, there is no information on the rate of internet penetration in Nigeria, no information on the demographic that utilize social media in Nigeria and there is no information on how these vary across States in Nigeria as the country has a wide gap in access to many social amenities, education, etc which can impact impact these and your findings.

3. One wonders how the authors were able to confirm that a study participant was a healthcare worker during the survey. It is important to transparently mention the extra checks that was put in place for this. This is critical as your study comapred the outcome among health workers and others.

4. Likewise, how were the authors able to confirm that a participant was 18 years or above. It is important to state was was done clearly or note this as a limitation of the study, as it will impact the interpretation of the study.

MINOR

1. There are several sentences in the manuscript that needs referencing. Please check the sentences starting in lines 70, 72, 110, 111 among others.

2. The authors states that NPHCDA is Nigeria "public health agency". The NCDC is Nigeria's puplic health agency while NPHCDA is the national primary healthcare agency. Please revise.

6. PLOS authors have the option to publish the peer review history of their article (what does this mean?). If published, this will include your full peer review and any attached files.

Reviewer #1: No

Reviewer #2: No

---

## [Author Response · Author response to Decision Letter 0]

29 Jun 2023

20 June 2023

PLOS ONE Editorial Office

Dear Editors:

On behalf of my co-authors, we thank the PLOS ONE editors for the opportunity to resubmit our revised manuscript entitled “Outcomes of a social media campaign to promote COVID-19 vaccination in Nigeria.” This revision provides responses to all editorial office comments and questions. Please see below for point-by-point responses. 

We certify that the manuscript has only been submitted to PLOS ONE for consideration. We also state that the project officer who was an employee of the funder at the time this project was initiated, Dr. Sohail Agha, participated in discussions about the study design and data collection. At the time of this manuscript preparation, Dr. Agha was no longer with the funder. Dr. Agha did participate in development of the manuscript as stated in the author roles section.

Please note that we will provide access to our data file upon request. At this time, we do not plan to provide accession numbers or DOIs. Data will be made available upon request.

If I may answer any other questions, please feel free to contact me at wdevans@gwu.edu.

Sincerely,

W. Douglas Evans, Ph.D.

Professor

 

Journal Requirements:

AUTHOR RESPONSE: WE HAVE FOLLOWED THE STYLE REQUIREMENTS AND CONFIRMED THAT THE MANUSCRIPT MEETS THE AUTHOR GUIDELINES.

AUTHOR RESPONSE: WE HAVE ENSURED THAT THEY MATCH. THE STUDY WAS FUNDED THROUGH INV-033413 FROM THE GATES FOUNDATION.

"Yes, Sohail Agha was a project officer and participated in meetings to design the study and provided input to the manuscript. The Principal Investigator, W. Douglas Evans, made all final study decisions."

AUTHOR RESPONSE: WE HAVE CLARIFIED THAT DR. AGHA PLAYED THE STATED ROLE THAT WE PROVIDED IN OUR INITIAL SUBMISSION AND WE INCLUDED THAT ROLE IN THE COVER LETTER (SEE ABOVE).

"This study was funded by the Bill & Melinda Gates Foundation (BMGF) grant INV-033413. We thank the foundation for its support of this study."

"Yes, Sohail Agha was a project officer and participated in meetings to design the study and provided input to the manuscript. The Principal Investigator, W. Douglas Evans, made all final study decisions."

AUTHOR RESPONSE: WE HAVE MADE THIS CORRECTION, AS REQUESTED. PLEASE NOTE THAT THE TEXT REGARDING SOHAIL AGHA SHOULD GO IN THE SECTION THAT DESCRIBES THE ROLE THAT THE FUNDER PLAYED IN THE STUDY. THE FUNDING WAS FROM THE BILL & MELINDA GATES FOUNDATION GRANT INV-033413.

AUTHOR RESPONSE: PLEASE NOTE THAT WE STATED WE WILL PROVIDE ACCESS TO OUR DATA UPON REQUEST, NOT THAT WE WILL PROVIDE REPOSITORY INFORMATION. THUS WE WILL NOT BE PROVIDING ACCESSION NUMBERS OR DOIs AT THIS TIME.

AUTHOR RESPONSE: WE HAVE STATED THE NAMES OF THE TWO COGNIZANT IRBs AND PROVIDED THE CONSENT METHOD IN THE METHODS SECTION.

7. Please upload a copy of Figure 1, to which you refer in your text on page 11. If the figure is no longer to be included as part of the submission please remove all reference to it within the text.

AUTHOR RESPONSE: THANK YOU, WE PROVIDED FIGURE 1 IN THE BODY OF THE MANUSCRIPT. WE HAVE ALSO ATTACHED IT AS A SEPARATE FILE IN THE RESUBMISSION.

Reviewer #1: Thank you for providing the opportunity to review this interesting study

The study focused on the impact of a social media campaign on COVID-19 vaccination that was premised on the principal of the theory of change (diffusion of innovations, social norm theory, and MOA framework model) among healthcare providers and the general population in Nigeria. The research evaluated the outcomes of the status of COVID-19 vaccine uptake, vaccine hesitancy, and pro-vaccination social norms among the participants recruited at 3 different time points over 10 months.

The study holds a good deal of merit for publication with following recommendations:

The introduction section does not sufficiently cover the topic mentioned in hand. Many studies can be cited from other LMIC, for instance:

1. https://doi.org/10.37723/jumdc.v12i2.624

2. https://doi.org/10.37723/jumdc.v13i3.762

AUTHOR RESPONSE: THANK YOU, WE HAVE EDITED THE INTRODUCTION, NOTING THAT THE FOCUS OF THE DISCUSSION IN THIS MANUSCRIPT IS ON DIGITAL HEALTH AND SOCIAL MEDIA RESEARCH.

The authors have targeted Facebook and Instagram to propagate social media campaigns. How this selection will ensure the generalizability of responses in the Nigerian context and not create a demographic bias? Please clarify what you mean by “the persuadable middle”

AUTHOR RESPONSE: IN THE PAPER, WE PROVIDE AN OPERATIONAL DEFINITION OF THE PERSUADABLE MIDDLE. WE ALSO NOTE THAT THE STUDY WAS AN EVALUATION OF AN INTERVENITON AND NOT A REPRESENTATIVE SAMPLE OF THE ENTIRE NIGERIAN POPULATION. WE HAVE ADDED CLARIFICATIONS IN THE METHODS SECTION.

The authors have chosen a mixed method study design. However, no detail or mention of the qualitative part of the survey. No mention of the stakeholders involved. How will you justify Line 99-101: “The evaluation employed mixed methods and comprised of a quantitative study conducted through social media-based surveys, a qualitative study of stakeholders, and a cost-effectiveness study”?

AUTHOR RESPONSE: WE NOTE AND HAVE CLARIFIED IN THE RESUBMISSION THAT THE QUALITATIVE PART OF THE STUDY IS REPORTED IN OTHER PUBLICATIONS.

The study mentions a quasi-experimental design with non-equivalent control and treatment groups. How will you justify the 6 states that you have chosen for treatment interventions? What was the rationale for not including the federal capital of Nigeria in the treatment group? Also, the terminology for mentioning the comparison or control group keeps on changing throughout the manuscript. Would have been better to keep it consistent for a better understanding

AUTHOR RESPONSE: WE CLARIFY THAT THE AUTHORS DID NOT SELECT THE LOCATIONS FOR THE INTERVENTION. THE STATES WERE SELECTED BY THE CAMPAIGN DEVELOPERS AND NIGERIAN GOVERNMENT.

Can the authors provide explicit detail on the social media campaign content that was used to assess the outcomes? Can they provide details on the interventional strategies used? How have they ensured that the findings obtained were merely based on the impact of social media campaigns and not on the other competing exposures?

AUTHOR RESPONSE: WHILE PROVIDING FULL DETAILS ON THE CAMPAIGN IS BEYOND THE SCOPE OF THIS ARTICLE, WE NOTED IN THE LIMITATIONS SECTION THAT OTHER CAMPAIGN EXPOSURE AND COMPETING INFLUENCES MAY AFFECT OUR FINDINGS.

The study mentions attrition bias as the limitation. However, can the authors provide a compelling rationale for the large loss of participants after the first follow-up?

AUTHOR RESPONSE: SUBSTANTIAL LOSS TO FOLLOW UP AFTER BASELINE IS COMMON IN LARGE SCALE SURVEY RESEARCH. THE LOSS TO FOLLOW UP IN 

THIS STUDY IS CONSISTENT WITH SIMILAR PREVIOUS RESEARCH.

To assess the primary outcome of COVID-19 vaccine status, please clarify does the fully vaccinated status was given with or without booster shots. How have you differentiated the 5Cs of vaccine hesitancy among participants with or without booster shots?

AUTHOR RESPONSE: VACCINATON STATUS WAS ASSESSED WITH A MULTI-COMPONENT QUESTION THAT ASKED ABOUT WHETHER THE INDIVIDUAL HAD HAD BOTH SHOTS OF A 2-SHOT REGIMEN (OR 1 SHOT IN A 1 SHOT REGIMEN). WE DID NOT ASK ABOUT BOOSTERS.

The discussion section lacks depth. Would have been better to draw comparisons of social media influence on COVID-19 vaccine hesitancy with other LMICs. The study demands a holistic explanation of the impact of social media campaigns in terms of the primary and secondary outcomes of your study.

AUTHOR RESPONSE: WE HAVE ADDED TO THE DISCUSSION SECTION.

Reviewer #2: In this manuscript, Evans and colleagues conducted a national-level quasi-experimental evaluation of a social media based COVID-19 vaccination promotion campaign across Nigeria's 36 states and the Federal Capital Territory. The authors purposively chose 6 States in the south-south, south-east, south-west, north-east, north-west and north-central region of Nigeria as "intervention" states where they ran online campaigns and compared them to Nigeria's other 30 States and the Federal Capital Territory as "comparator States".

The study used a potentially novel methodology for data collection. It is also evident that a lot of work and collbaorative effort has gone in to ensure this study was succesfully completed. That being said, there are some important issues that I strongly recommend that the authors address to improve the overall quality of the manuscript. Ihave lsited below as major and minor issues.

MAJOR

1. I would like to address a significant concern regarding the absence of local authors in the list of authors. In recent years, the global health community has increasingly emphasized the importance of equitable and inclusive collaborations, particularly in research conducted in low- and middle-income countries. It is crucial to ensure that local researchers, scientists, and collaborators from the host country are included as authors, as their perspectives, insights, and expertise are vital in conducting research that accurately reflects the local context and addresses the specific health needs of the population. I note that in line 104, the authors state that were invloved in designing the campaign. I also note that the study has IRB from Nigeria's national primary health care agency (not "public health agency" as stated by the authors") which I assume was led by local collaborators too. To make matthers worse, your acknowledgment section only mentioned the study funder and not even a mention of a lcoal collaborator whom have 

been referred to in the body of the manuscript as creators of the campaign and delivered the campaigns.

Therefore, I strongly recommend that you consider including local authors who have contributed significantly to the study design, data collection, analysis, interpretation, or manuscript preparation. This would enhance the credibility and relevance of the research findings, foster meaningful partnerships, and support the broader goal of promoting local ownership and sustainability in research endeavors. I understand that there may be various reasons for the absence of local authors in the current version of the manuscript. If there are limitations or challenges in involving local authors, I encourage you to provide a clear and transparent explanation in the manuscript, outlining the efforts made to engage local collaborators and the reasons for their exclusion as authors. By addressing this issue, your study will not only comply with emerging guidelines and expectations in global health research but also contribute to the advancement of ethical research practices, respectful collaborations, and knowledge co-creation in LMIC settings.

AUTHOR RESPONSE: WE ACKNOWLEDGE AND AGREE WITH THE IMPORTANT ISSUE RAISED BY THE REVIEWER. PLEASE NOTE THAT TWO OF THE CO-AUTHORS, DR. KHADI NDIAYE AND MS. SELINAM AKABA, ARE WEST AFRICAN (FROM SENEGAL AND NIGERIA). THERE WAS ONE OTHER LOCAL COLLABORATOR, MR. IFEANYI NSOFOR, WHO WORKED ON THE RESEARCH AND WE HAVE ADDED HIM AS A CO-AUTHOR AND NOTED HIS ROLES IN THE AUTHOR ROLES SECTION. 

2. The manuscript methods sections lacked information on the local context in Nigeria which would be useful in contextualizing the findings. For example, there is no information on the rate of internet penetration in Nigeria, no information on the demographic that utilize social media in Nigeria and there is no information on how these vary across States in Nigeria as the country has a wide gap in access to many social amenities, education, etc which can impact impact these and your findings.

AUTHOR RESPONSE: WE HAVE ADDED DETAILS ON THE RELEVANT TECHNOLOGY TO THIS PROJECT, AND PROVIDED A CITATION ON SOCIAL MEDIA USE IN NIGERIA.

3. One wonders how the authors were able to confirm that a study participant was a healthcare worker during the survey. It is important to transparently mention the extra 

checks that was put in place for this. This is critical as your study comapred the outcome among health workers and others.

AUTHOR RESPONSE: WE ASKED SCREENING QUESTIONS TO CONFIRM HEALTHCARE WORKER STATUS AND ASKED FOR THE SPECIFIC JOB ROLE (E.G., DOCTOR, NURSE, ETC.). WE ADDED THIS TO THE METHODS SECTION.

4. Likewise, how were the authors able to confirm that a participant was 18 years or above. It is important to state was was done clearly or note this as a limitation of the study, as it will impact the interpretation of the study.

AUTHOR RESPONSE: LIKEWISE, WE ASKED SCREENING QUESTIONS.

MINOR

1. There are several sentences in the manuscript that needs referencing. Please check the sentences starting in lines 70, 72, 110, 111 among others.

2. The authors states that NPHCDA is Nigeria "public health agency". The NCDC is Nigeria's puplic health agency while NPHCDA is the national primary healthcare agency. Please revise.

AUTHOR RESPONSE: WE THANK THE REVIEWER AND HAVE MADE REVISIONS.

---

## [Decision Letter · Decision Letter 1]

12 Jul 2023

PONE-D-23-15997R1Outcomes of a social media campaign to promote COVID-19 vaccination in NigeriaPLOS ONE

 Dear Dr. Evans,

Thank you for submitting your manuscript to PLOS ONE. After careful consideration, we feel that it has merit but does not fully meet PLOS ONE’s publication criteria as it currently stands. Therefore, we invite you to submit a revised version of the manuscript that addresses the points raised during the review process. Please ensure that your decision is justified on PLOS ONE’s publication criteria and not, for example, on novelty or perceived impact.

We look forward to receiving your revised manuscript.

Kind regards,

Naeem Mubarak, PhD

Academic Editor

PLOS ONE

Additional Editor Comments:

The study merits publication with improvements as advised in the reviewer's comments

Reviewers' comments:

Reviewer's Responses to Questions

**Comments to the Author**

1. If the authors have adequately addressed your comments raised in a previous round of review and you feel that this manuscript is now acceptable for publication, you may indicate that here to bypass the “Comments to the Author” section, enter your conflict of interest statement in the “Confidential to Editor” section, and submit your "Accept" recommendation.

Reviewer #1: (No Response)

Reviewer #2: All comments have been addressed

2. Is the manuscript technically sound, and do the data support the conclusions?

Reviewer #1: Partly

Reviewer #2: Yes

3. Has the statistical analysis been performed appropriately and rigorously? 

Reviewer #1: Yes

Reviewer #2: Yes

4. Have the authors made all data underlying the findings in their manuscript fully available?

Reviewer #1: Yes

Reviewer #2: Yes

5. Is the manuscript presented in an intelligible fashion and written in standard English?

Reviewer #1: Yes

Reviewer #2: Yes

6. Review Comments to the Author

Reviewer #1: Thank you for providing the opportunity to review the revised manuscript

The comments have not been completely addressed

It is necessary to update the introduction section. No changes suggested previously were made. You may use the following studies to cover the insufficient details of the topic in hand.

1. https://doi.org/10.37723/jumdc.v12i2.624

2. https://doi.org/10.37723/jumdc.v13i3.762

The discussion section was edited but with no details on the comparison of your findings with other countries as recommended previously.

Can the authors explicitly mention the sample size calculation separately for both the HCP and the general population? What population size you used to reach your target sample size for both? Is your survey response sufficient to meet your target sample size for both populations? Please clarify.

Reviewer #2: I have no further comments. All required questions have been answered. I have no concerns with publication ethics

7. PLOS authors have the option to publish the peer review history of their article (what does this mean?). If published, this will include your full peer review and any attached files.

Reviewer #1: No

Reviewer #2: No

---

## [Author Response · Author response to Decision Letter 1]

14 Jul 2023

12 July 2023

PLOS ONE Editorial Office

Dear Editors:

On behalf of my co-authors, we thank the PLOS ONE editors for the opportunity to resubmit our submit a second revision of the manuscript entitled “Outcomes of a social media campaign to promote COVID-19 vaccination in Nigeria.” This revision provides responses to all editorial office comments and questions. Note that revised manuscript content is highlighted in yellow. Please see below for point-by-point responses. 

We certify that the manuscript has only been submitted to PLOS ONE for consideration. We also state that the project officer who was an employee of the funder at the time this project was initiated, Dr. Sohail Agha, participated in discussions about the study design and data collection. At the time of this manuscript preparation, Dr. Agha was no longer with the funder. Dr. Agha did participate in development of the manuscript as stated in the author roles section.

Please note that we will provide access to our data file upon request. At this time, we do not plan to provide accession numbers or DOIs. Data will be made available upon request.

If I may answer any other questions, please feel free to contact me at wdevans@gwu.edu.

Sincerely,

W. Douglas Evans, Ph.D.

Professor

 

Reviewer #1: Thank you for providing the opportunity to review the revised manuscript

The comments have not been completely addressed

It is necessary to update the introduction section. No changes suggested previously were made. You may use the following studies to cover the insufficient details of the topic in hand.

1. https://doi.org/10.37723/jumdc.v12i2.624

2. https://doi.org/10.37723/jumdc.v13i3.762

AUTHOR RESPONSE: WHERE APPROPRIATE, WE EDITED THE INTRODUCTION TO NOTE ISSUES SURROUNDING MIS- AND DISINFORMATION AND ADDED A CITATION.

The discussion section was edited but with no details on the comparison of your findings with other countries as recommended previously.

AUTHOR RESPONSE: WE ADDED A NOTE ABOUT THE GLOBAL CONTEXT OF RELEVANT COVID-19 VACCINATION RESEARCH IN LMICS.

Can the authors explicitly mention the sample size calculation separately for both the HCP and the general population? What population size you used to reach your target sample size for both? Is your survey response sufficient to meet your target sample size for both populations? Please clarify.

AUTHOR RESPONSE: WE ADDED A NOTE THAT SAMPLE SIZE WAS ESTIMATED TO PROVIDE SUFFICIENT SAMPLE BOTH TO DETECT A CHANGE IN HCP AND GENERAL POPULATION VACCINATION RATES.

---

## [Decision Letter · Decision Letter 2]

15 Aug 2023

Outcomes of a social media campaign to promote COVID-19 vaccination in Nigeria

PONE-D-23-15997R2

Dear Dr. W. Douglas Evans,

We’re pleased to inform you that your manuscript has been judged scientifically suitable for publication and will be formally accepted for publication once it meets all outstanding technical requirements.

Comments from PLOS Editorial Office: We note that during previous rounds of peer review one or more reviewers has recommended that you cite specific previously published works. As always, we recommend that you please review and evaluate the requested works to determine whether they are relevant and should be cited. It is not a requirement to cite these works, and if you assess that any such requested citations are not sufficiently relevant to this study, you may remove them during preparation of the final manuscript files. We appreciate your attention to this request.

Kind regards,

Naeem Mubarak, PhD

Academic Editor

PLOS ONE

Additional Editor Comments (optional):

The manuscript holds a good deal of merit for publication with no further revisions required and may be accepted.

Reviewers' comments:

Reviewer's Responses to Questions

**Comments to the Author**

1. If the authors have adequately addressed your comments raised in a previous round of review and you feel that this manuscript is now acceptable for publication, you may indicate that here to bypass the “Comments to the Author” section, enter your conflict of interest statement in the “Confidential to Editor” section, and submit your "Accept" recommendation.

Reviewer #1: All comments have been addressed

Reviewer #3: All comments have been addressed

2. Is the manuscript technically sound, and do the data support the conclusions?

Reviewer #1: Yes

Reviewer #3: Yes

3. Has the statistical analysis been performed appropriately and rigorously? 

Reviewer #1: Yes

Reviewer #3: Yes

4. Have the authors made all data underlying the findings in their manuscript fully available?

Reviewer #1: Yes

Reviewer #3: Yes

5. Is the manuscript presented in an intelligible fashion and written in standard English?

Reviewer #1: Yes

Reviewer #3: Yes

6. Review Comments to the Author

Reviewer #1: Thankyou for providing me the opportunity to review this revised manuscript. I have no further concerns. All the comments have been addressed fully. The study now holds a good deal of merit for publication with improved quality.

Reviewer #3: All the queries have been addressed by the author. I do not have any further concerns regarding publication of this article. Good Luck!

7. PLOS authors have the option to publish the peer review history of their article (what does this mean?). If published, this will include your full peer review and any attached files.

Reviewer #1: No

Reviewer #3: No

---

## [Editor Report · Acceptance letter]

5 Sep 2023

PONE-D-23-15997R2 

Outcomes of a social media campaign to promote COVID-19 vaccination in Nigeria 

Dear Dr. Evans:

I'm pleased to inform you that your manuscript has been deemed suitable for publication in PLOS ONE. Congratulations! Your manuscript is now with our production department. 

Kind regards, 

on behalf of

Dr Naeem Mubarak 

Academic Editor

PLOS ONE